# HEURISTICS FOR IMAGE GENERATION FROM SCENE GRAPHS

**Subarna Tripathi, Anahita Bhiwandiwalla, Alexei Bastidas & Hanlin Tang**
Intel AI Lab
Santa Clara, USA
{subarna.tripathi,anahita.bhiwandiwalla,alexei.bastidas,hanlin.tang}@intel.com

## ABSTRACT

Generating realistic images from scene graphs requires neural networks to be able to reason about object relationships and compositionality. Learning a sufficiently rich representation to facilitate this reasoning is challenging due to dataset limitations. Synthetic scene graphs from COCO only have basic geometric relationships, and Visual Genome scene graphs are replete with missing relations or mislabeled nodes. Existing scene graph to image models have two stages: (1) a scene composition stage, and an (2) image generation stage. In this paper, we propose two methods to improve the intermediate representation of these stages. First, we use visual heuristics to augment relationships between pairs of objects. Second, we introduce a graph convolution-based network to generate a scene graph context representation that enriches the image generation. These contributions significantly improve the scene composition (relation score of 59.8% compared to 51.2%) and image generation (74% versus 64% in mean relation opinion score). Introspection shows that these heuristics are particularly effective in learning differentiated representations for scenes with multiple instances of the same object category. Obtaining accurate and complete scene graph annotations is costly, and our use of heuristics and prior structure to enhance intermediate representations allows our model to compensate for limited or incomplete data.

## 1 INTRODUCTION

The generation of realistic scenes marks an important challenge for neural networks, with recent advancements enabling synthesizing high-resolution images, even when they are conditioned on class labels (Mirza & Osindero, 2014), captions (Reed et al., 2016), or latent dimensions (Karras et al., 2018). However, the ability to interpret object sizes, relationships, and composition to synthesize realistic scenes still eludes neural networks.

Johnson et al (Johnson et al., 2018) recently proposed generating images from scene graphs. Scene graphs are a structured representation, with objects as nodes, and edges marking the semantic relationship between objects. The scene graph is used to generate a scene composition in the form of a segmentation mask, which a network consumes to generate a realistic RGB image. This method yields significantly improved generated images, but notably struggles with cluttered or small objects.

Several dataset limitations render this problem challenging. High quality scene graph annotations are costly, and existing datasets such Visual Genome have scene graphs that are not complete, with noisy or missing relations and nodes. Researchers have resorted to generating synthetic scene graphs from COCO-stuff datasets, but those graphs are limited to simple geometric relationships (above, below, left, right). These dataset limitations restrict the richness of the learned representations that are required for accurate scene composition and image generation.

In this paper, we employ heuristics and prior structure to learn more complex representations, and show that these contributions significantly improve overall performance, and lead to more differentiated representations for similar objects in the same scene. First, we use linear perspective based heuristics to augment the relationship between objects. Second, we introduce a scene context network to provide context feature representations to both the generator and discriminator, which incentives compliance of the generated images to the scene graph. Generating images from scene

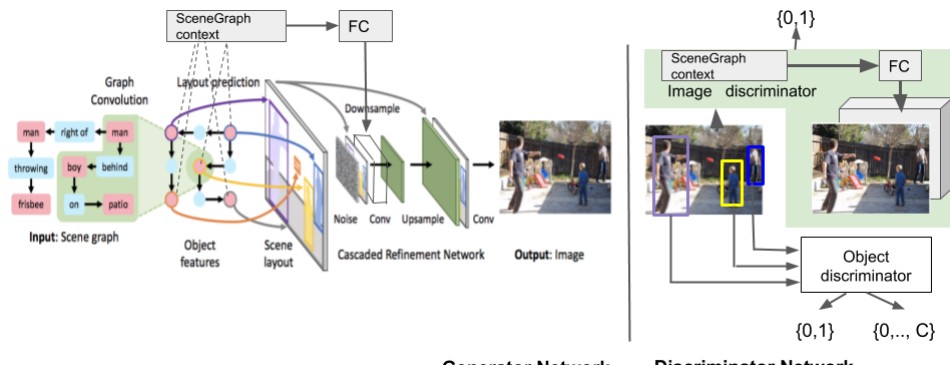

Figure 1: Model has a scene layout stage, and an image generation stage (Cascaded Refinement Network). We use a scene graph context network to augment the representation for the generator as well as the discriminator. Graphics partially adapted from Johnson et al. (2018)

graph is a relatively new task without well-defined metrics. We introduce novel evaluation metrics more suited for this task: the relation score and mean opinion relation score (MORS), both of which measure the compliance of generated images to the scene graph. Our proposed model significantly outperforms the non-heuristic approach on both quantitative and qualitative metrics.

## 2 METHOD

Given a scene graph consisting of objects and their relationships, our model constructs realistic image corresponding to the scene graph. Our framework is built upon (Johnson et al., 2018). Briefly, the scene graph is converted into object embedding vectors from a Graph Convolution Neural Network (GCNN), which are then used to predict bounding boxes and segmentation masks for each object (the "Scene Composition" stage). This mask layout is an intermediate between the graph and image domains. Finally, a Cascade Refinement Network (Chen & Koltun, 2017) generates the image. See Figure 1.

Converting from the graph to the image domain requires learning intermediate representations that combine structured information from the scene graph with predicted positional information. To enhance these representations when the input dataset has limited or incomplete scene graphs, we employ three main contributions:

**Heuristic-based Data Augmentation.** We used heuristics to augment the scene graphs with new spatial relations that induce a richer learned representation. We quasi-exhaustively determined the depth order between objects from observers' viewpoint. For 2D images, determining this order is non-trivial. We utilize linear perspective based heuristics instead for augmenting spatial relationship vocabulary. We provide the details in the experiments section.

**Scene graph context.** In the existing work, a segmentation mask layout serves as the intermediate representation between the input graph domain and the output image domain. However, this representation loses any context from the scene graph, reducing the quality of the image generation stage. In our model, we add a scene graph context network that pools feature generated from the graph convolution. These pooled context features are then passed to a fully-connected layer that generates embeddings that are provided to the image generation stage, as well as to an adversarial loss. In this way, we directly augment the intermediate representation with scene graph context embeddings.

**Matching-aware Loss.** To further encourage the model to generate images that match input scene graph descriptions, we employ a matching-aware loss (Reed et al., 2016; Hong et al., 2018). We construct mismatched triplets (layout, graph embedding, image) as fake examples during adversarial training.

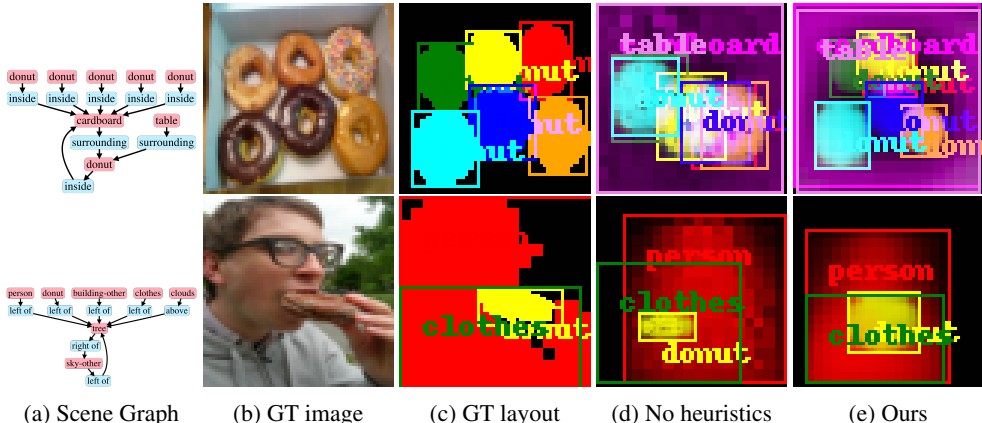

| (a) Scene Graph | (b) GT image | (c) GT layout | (d) No heuristics | (e) Ours |

Figure 2: Layout generation: Model learned without the data augmentation notably struggles with placing objects o apart when the input scene graph does not specify explicit relationships between those (Row 1). Heuristic based data augmentation significantly outperform the layout prediction for incomplete scene graph inputs. Data augmentation helps is learning better inter-class object placements (e.g person and clothes) (Row 2).

## 3 EXPERIMENTS

We train our model to generate $64 \times 64$ images on the Visual Genome (Krishna et al., 2016) and COCO-Stuff (Caesar et al., 2018) datasets. In our experiments we aim to show that the generated images look realistic and they respect the objects and relationships of the input scene graph.

### 3.1 DATASETS

**COCO**: We performed experiments on the 2017 COCO-Stuff dataset Caesar et al. (2018), which augments a subset of the COCO dataset Lin et al. (2014) with additional stuff categories. Similar to Johnson et al. (2018), we first used thing and stuff annotations to construct synthetic scene graphs based on the 2D image coordinates of the objects, encoding six mutually exclusive geometric relationships: 'left of', 'right of', 'above', 'below', 'inside', and 'surrounding'.

In the augmentation process, we exploit the 'thing' and 'stuff' annotations. We encode two heuristics based relationships: 'behind' and 'in front of' between two spatially overlapping 'things'. Among overlapping 'things' A and B, A is 'in front of' B if the bottom boundary of A's bounding box is closer to the image's bottom edge. Additionally, 'on' and 'under' relationships between overlapping 'thing' and 'stuff' are imposed. A 'thing' is always 'on' the 'stuff' from viewer's perspective. Significant number of COCO images contain non-iconic object pose.

**Visual Genome**: We experimented on Visual Genome [26] version 1.4 (VG) which comprises of 108,077 images annotated with scene graphs. We used the pre-processing described in Johnson et al. (2018), resulting in a training set with 178 object and 45 relationship types. We ignored small objects, and only selected images with object counts between 3 and 30 and at least one relationship. Visual Genome does not provide segmentation masks.

### 3.2 QUANTITATIVE RESULTS: LAYOUT PREDICTION

We first compare model performance at the intermediate stage of layout prediction. For COCO, we compare the predicted layout with the ground truth using both Intersection-over-Union (IoU) but also a novel Relation Score. IoU does not capture spatial relationships, so we define Relation Score as the fraction of spatial relationships ('above', 'below', etc.) between objects that are satisfied in the predicted layout. By applying our heuristic-based data augmentation, we significantly improve the performance of the model (Table 1) for both metrics.

Input scene graphs are often incomplete. We observe that without direct edges between objects of same class, the underlying representations are not well differentiated. This leads to the layout network placing all the objects in similar locations. For example, see the top row in Figure 2. Without heuristics (part d), the six donuts are all overlapping, where as with the heuristics, the model is able to unclutter the scene.

| Metric | No DA | DA |
|---|---|---|
| Avg IOU | 0.459 | **0.510** |
| Relation Score | 0.512 | **0.598** |

Table 1: Relationship compliance. Relation score (the higher the better) on COCO stuff test set. DA stands for data augmentation.

| Study | Johnson *et al* | **Ours** |
|---|---|---|
| AB-X (Caption) | 42.6% | **57.4%** |
| AvB | 42.7% | **58.3%** |

Table 2: Visual Genome qualitative study results. Our model outperformed Johnson when workers were asked to select which image was more realistic (AvB), and which image better matched a provided pseudo-caption (AB-X).

## 3.3 QUALITATIVE METRICS

To evaluate the effect of enriching the intermediate representation with scene graph context embeddings, we used Amazon Turk to rate the generated images. We asked Turkers three questions: which image was a closer match to the caption (AB-X), which image was more realistic (AvB), and whether a given relation (e.g. sheep covered in grass) was true in the image (the Mean Opinion Relation Score). Results for first two tasks are shown in Table 2. To compute the MORS, we selected single image-relationship pairs, and asked workers to rate whether the relationship is true in the image (Figure 3). MORS is then defined as the fraction of tested relationships that were found present in the generated image.

Our model outperformed the baseline model in all of these qualitative metrics (Tables 1-3), demonstrating the importance of using heuristic data augmentation, as well as better methods of representation learning at the graph-to-image interface. Importantly, our model was significantly better on non-spatial relationships such as semantic (0.78 vs 0.60) or possessive (0.80 vs 0.62). The scene context network includes semantic embeddings, which may be contributing to this improvement.

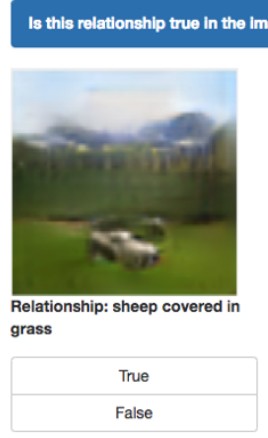

Figure 3: MORS data collection

| Relation category | JJ | **Ours** |
|---|---|---|
| Semantic | 0.60 | **0.78** |
| Geometric | 0.64 | **0.68** |
| Possessive | 0.62 | **0.80** |
| Miscellaneous | 0.78 | **0.86** |
| Overall MORS | 0.64 | **0.74** |

Table 3: Mean Opinion Relation score (MORS) on 100 random images and relationship pairs generated from the colored scene graphs in Visual Genome test set. The score is broken by relation category. Each image was rated by five workers. IoU corresponds to all predicted boxes in test set

## 4 CONCLUSION

Progress in scene-graph related tasks, such as the image generation task studied here, has been slow for two main reasons. First, datasets such as Visual Genome are hampered by incomplete and incorrect scene graph annotations, or synthetic datasets such as COCO-stuff with relatively simple spatial relationships. Second, lack of metrics designed to measure scene graph compliance. Our proposed heuristics and prior structure to enhance intermediate representations allows our model to compensate for limited or incomplete data. We also introduced Relation Score and MORS metrics that measure compliance at the scene layout and generated images stages, respectively.

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
