# OpenReview forum: "Heuristics for Image Generation from Scene Graphs"
_ICLR.cc/2019/Workshop/LLD — LLD 2019_

### Official Review · AnonReviewer2 · 2019-04-07
**Article uses "depth ordering from observers viewpoint" between objects as heuristic to improve generation of realistic images from scene graphs by "augmenting the scene graphs". Article seems very vague on this very contribution.**

**Rating:** 1
**Confidence:** 2

**Review:**

The article claims to improve the scene graph to image generation task by augmenting the scene graph dataset with ordering of object information. The article also claims to have contributed a new graph CNN "We use a scene graph context network to augment the representation for the generator as well as the discriminator".

Both of these constributions are vague and incompletely defined.

Though the article contains some experiments and a brief descroption of the ordering heuristic, while no mention of the Graph CNN, it is difficult to ascertain the contribution.

---

### Official Review · AnonReviewer1 · 2019-04-08
**Extensive evaluation of a data augmentation technique for image generation from scene graphs.**

**Rating:** 4
**Confidence:** 2

**Review:**

This paper deals with the problem of image generation from scene graphs, building on Johnson et al ( Image generation from scene graphs, 2018). There are three main contributions in the paper:

1. a data augmentation scheme that employs heuristics to add fine-grained annotations of spatial relationships between pairs of objects in the scene, e.g., "on top of", "left of", "behind", etc.

2. A graph neural network that adds context on top of object segmentation masks, to maintain information about the relationships between objects.

3. A new evaluation metric that measures the compliance of the generated images to the (augmented) ground truth scene graph, as a fraction of the satisfied spatial relationships between objects in the ground truth.

The experiments are expensive, including even a perceptual study using amazon turkers, and the results show noticeably improved performance, compared to the baseline, both in terms of IOU and the new proposed metric (MORS). I have a question/remark though: when the authors  describe the heuristics they used to augment the data, they claim that " A is ’in front of’ B if the bottom boundary of A’s bounding box is closer to the image’s bottom edge". I don't think this is true in the case where the bounding box of B is fully contained in the bounding box of A, in which case B is in front of A.

---

### Decision · Program_Chairs · 2019-04-09
**Acceptance Decision**

**Decision:**

Accept

**Comment:**

The authors propose two contributions a) data augmentation techniques for scene graph to image generation as well as b) a new mechanism for the scene graph to image generation that maintains context using a GCNN.

Pros:
- The augmentation strategy makes sense and is reasonably illustrated
- Improve on highly relevant problem that is not well solved or easily evaluated as the authors mentioned
-The metareviewer and R1 appreciate the perceptual studies with amazon turk.
-The first contribution is well in keeping with the theme of this workshop.

Cons:
	-As mentioned by R2 more details should really be included at least in the appendix. E.g. any major differences to the JJ pipeline such as size and form of the Graph CNN context rep. The authors also should cite the graphic copied from Johnson et al.
- No ablations to show the effect of the different contributions compared to JJ (it is not completely clear whether the gain comes from the augmentation, use of the context GCNN or from architecture/other changes to JJ pipeline).
-It should be made more clear if the scene augmentations are also used for the evaluation scene graphs and if so whether the JJ model also sees the same augmented scenes at evaluation.

Overall this paper handles a very difficult and challenging problem and both contributions as well as the suggested evaluations are substantial.